# MEASURING CALIBRATION IN DEEP LEARNING

## ABSTRACT

Overconfidence and underconfidence in machine learning classifiers is measured by *calibration*: the degree to which the probabilities predicted for each class match the accuracy of the classifier on that prediction. We propose new measures for calibration, the Static Calibration Error (SCE) and Adaptive Calibration Error (ACE). These measures taks into account every prediction made by a model, in contrast to the popular Expected Calibration Error.

## 1 INTRODUCTION

The reliability of a machine learning model's confidence in its predictions is critical for high risk applications, such as deciding whether to trust a medical diagnosis prediction (Crowson et al., 2016; Jiang et al., 2011; Raghu et al., 2018). One mathematical formulation of the reliability of confidence is calibration (Murphy & Epstein, 1967; Dawid, 1982).

Intuitively, for class predictions, calibration means that if a model assigns a class with 90% probability, that class should appear 90% of the time.

Calibration is not directly measured by proper scoring rules like negative log likelihood or the quadratic loss. The reason that we need to assess the quality of calibrated probabilities is that presently, we optimize against a proper scoring rule and our models often yield uncalibrated probabilities.

Recent work proposed Expected Calibration Error (ECE; Naeini et al., 2015), a measure of calibration error which has lead to a surge of works developing methods for calibrated deep neural networks (e.g., Guo et al., 2017; Kuleshov et al., 2018). In this paper, we show that ECE has numerous pathologies, and that recent calibration methods, which have been shown to successfully recalibrate models according to ECE, cannot be properly evaluated via ECE.

Calibration (and uncertainty quantification generally) is critical in autonomous vehicles, the exploration phase of may algorithms, medical applications, and many more safety-critical applications of machine learning. A suite of recent papers (e.g., Lee et al., 2017; Vaicenavicius et al., 2019; Thulasidasan et al., 2019; Kumar et al., 2018; Guo et al., 2017; Seo et al., 2019) use ECE to validate their models' calibration. We identify concerns with that methodology. As this metric has become the default choice for measuring calibration in industry and for researchers, we expect our criticism to change the decisions of anyone attempting to train well-calibrated classification models. We recommend that rather than using Expected Calibration Error, practitioners use Static Calibratinon Error or Adaptive Calibration.

## 2 BACKGROUND & RELATED WORK

### 2.1 MEASUREMENT OF CALIBRATION

Assume the dataset of features and outcomes $\{(x, y)\}$ are i.i.d. realizations of the random variables $X, Y \sim \mathbb{P}$. We focus on class predictions. Suppose a model predicts a class $y$ with probability $\hat{p}$. The model is *calibrated* if $\hat{p}$ is always the true probability. Formally,

$$\mathbb{P}(Y = y \mid \hat{p} = p) = p$$

for all probability values $p \in [0, 1]$ and class labels $y \in \{0, \dots, K - 1\}$. The left-hand-side denotes the true data distribution's probability of a label given that the model predicts $\hat{p} = p$; the right-hand-side denotes that value. Any difference between the left and right sides for a given $p$ is known as *calibration error*.

**Expected Calibration Error (ECE).** To approximately measure the calibration error in expectation, ECE discretizes the probability interval into a fixed number of bins, and assigns each predicted probability to the bin that encompasses it. The calibration error is the difference between the fraction of predictions in the bin that are correct (accuracy) and the mean of the probabilities in the bin (confidence). Intuitively, the accuracy estimates $\mathbb{P}(Y = y \mid \hat{p} = p)$, and the average confidence is a setting of $p$. ECE computes a weighted average of this error across bins:

$$\text{ECE} = \sum_{b=1}^{B} \frac{n_b}{N} \left| \text{acc}(b) - \text{conf}(b) \right|,$$

where $n_b$ is the number of predictions in bin $b$, $N$ is the total number of data points, and $\text{acc}(b)$ and $\text{conf}(b)$ are the accuracy and confidence of bin $b$, respectively. ECE as framed in Naeini et al. (2015) leaves ambiguity in both its binning implementation and how to compute calibration for multiple classes. In Guo et al. (2017), they bin the probability interval $[0, 1]$ into equally spaced subintervals, and they take the maximum probability output for each datapoint (i.e., the predicted class's probability). We use this for our ECE implementation.

## 2.2 PROPER SCORING RULES & OTHER MEASURES OF CALIBRATION.

Many classic methods exist to measure the accuracy of predicted probabilities. For example, the Brier score (quadratic loss) measures the mean squared difference between the predicted probability and the actual outcome (Gneiting & Raftery, 2007). This score can be shown to decompose into a sum of metrics, including calibration error. In practice, training against this proper scoring rule, like training against log likelihood, does not guarantee a well calibrated model. The Hosmer-Lemeshow test is a popular hypothesis test for assessing whether a model's predictions significantly deviate from perfect calibration (Hosmer & Lemesbow, 1980). The reliability diagram provides a visualization of how well-calibrated a model is (DeGroot & Fienberg, 1983). Adaptivity was explored by (Nguyen & O'Connor, 2015) and applied in (Hendrycks et al., 2018). Kuleshov et al. (2018) extends ECE to the regression setting. Unlike the Brier score and Hosmer-Lemeshow, we'd like the metric to be scalar-valued in order to easily benchmark methods, and to only measure calibration.

## 2.3 POST-PROCESSING METHODS

### 2.3.1 STANDARD APPROACHES FOR MULTICLASS CLASSIFICATION

One common approach to calibration is to apply post-processing methods to the output of any classifier without retraining. The two most popular post-processing methods are the parametric approach of Platt scaling (Platt et al., 1999) and the non-parametric approach of isotonic regression (Zadrozny & Elkan, 2002).

Platt scaling fits a logistic regression model to the logits of a classifier, on the validation set, which can be used to compute calibrated predictions at test time. The original formulation of Platt scaling for neural networks (Niculescu-Mizil & Caruana, 2005) involves learning scalar parameters $a, b \in \mathbb{R}$ on a held-out validation set and then computing calibrated probabilities $\hat{p}_i$ given the uncalibrated logits vector $z_i$ on the test set as $\hat{p}_i = \sigma(az_i + b)$. These parameters are typically estimated by minimizing the negative log likelihood.

Platt scaling can be extended to the multiclass setting by considering higher-dimensional parameters. In *matrix scaling* $a$ is replaced with $W \in \mathbb{R}^{K*K}$ while we consider $b \in \mathbb{R}^K$. As for *vector scaling*; $W \in \mathbb{R}^K$. Calibrated probabilities for either of these extensions can then be computed as

$$\hat{p}_i = \max_k \sigma(Wz_i + b).$$

An even simpler extension is *temperature scaling* (Guo et al., 2017) which reduces the set of regression parameters to the inverse of a single scalar $T > 0$ such that

$$\hat{p}_i = \max_k \sigma(z_i/T).$$

On the other hand, isotonic regression is a common non-parametric processing method that finds the stepwise-constant non-decreasing (isotonic) function $f$ that best fits the data according to a

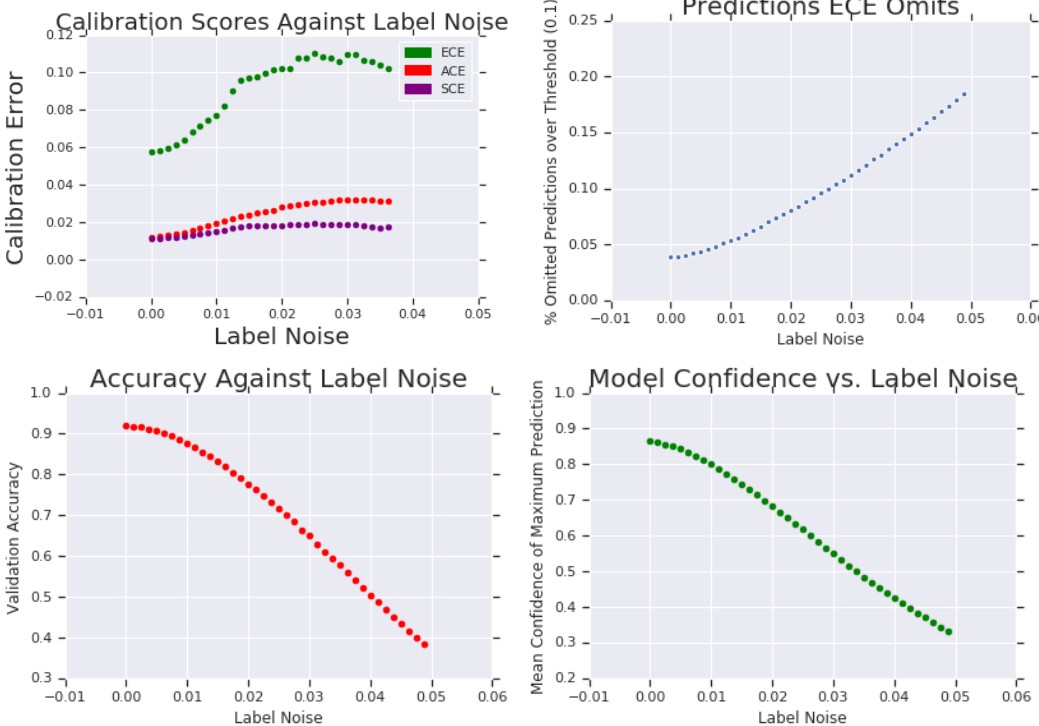

Figure 1: **Top Left:** Calibration measured through three metrics on MNIST with randomly assigned labels, with the fraction of randomly assigned labels on the x axis (all others labels are correct). Models (here, multinomial logistic regression) are re-trained at each noise level. The base level of calibration error will be different, as ACE and SCE account for all predictions, while ECE only accounts for the prediction with maximum confidence for each sample. These predictions have different average calibration errors.**Top Right:** Often, important predictions (confidence > 0.1) are not captured in the calibration error by ECE. As models trained with more label noise exhibit more uncertainty, that number increases as the amount of label noise increases. **Bottom:** As label noise increases, model output certainty and accuracy decrease.

mean-squared loss function $\sum_i (f(p_i) - y_i)^2$ where $p_i$ are the uncalibrated probabilities and $y_i$ the labels.

The standard approach for extending isotonic regression to the multiclass setting is to break the problem into many binary classification problems (e.g. one-versus-all problems), to calibrate each problem separately, and then to combine the calibrated probabilities (Zadrozny & Elkan, 2002).

## 3 ISSUES WITH CALIBRATION METRICS

### 3.1 NOT COMPUTING CALIBRATION ACROSS ALL PREDICTIONS

Expected Calibration Error was crafted to mirror reliability diagrams, which are structured around binary classification such as rain vs not rain (DeGroot & Fienberg, 1983) (Figure 1). A consequence is that the error metric is reductive in a multi-class setting. In particular, ECE is computed using only the predicted class's probability, which implies the metric does not assess how accurate a model is with respect to the $K - 1$ other probabilities.

We examine increasing the importance of omitted predictions via label noise. As label noise increases, trained models become less confident in their predictions. Secondary and tertiary predictions are correct with higher frequency, and so for many samples the prediction with the correct label isn't included in ECE's measure of calibration error. We find that ECE becomes a worse approximation of

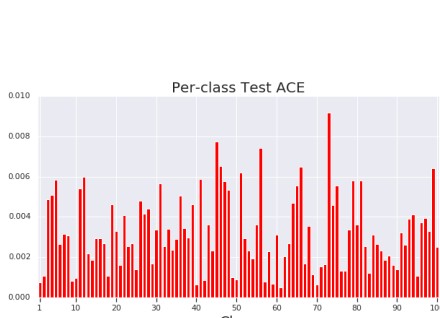 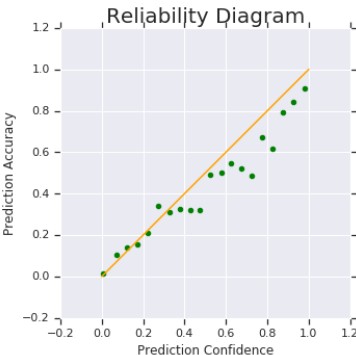

Figure 2: **Left:** Per-class Adaptive Calibration Error (ACE) for a trained 110-layer ResNet on the CIFAR-100 test set. We observe that calibration error is non-uniform across classes, which is difficult to express when measuring error only on the maximum probabilities. **Right:** Reliability Diagrams are an interpretable visualization of calibration error. This is an example of the reliability of a slightly overconfident CNN trained on Fashion-MNIST.

the calibration error as relevant predictions not included in the ECE metric become common (Figure 1).

In Figure 2, we observe that calibration error, when measured in a per-class manner, is non-uniform across classes. Guaranteeing calibration for all class probabilities is an important property for many applications. For example, a physician may care deeply about a secondary or tertiary prediction of a disease. Decision costs along both ethical and monetary axes are non-uniform, and thus clinical decision policies necessitate a comprehensive view of the likelihoods of potential outcomes (e.g., Tsoukalas et al., 2015). Alternatively, if a true data distribution exhibits high data noise (also known as high aleatoric uncertainty, or weak labels), then we'd like models to be calibrated across all predictions as no true class label is likely.

### 3.2 FIXED CALIBRATION RANGES

One major weakness of evenly spaced binning metrics is caused by the dispersion of data across ranges. In computing ECE, there is often a large leftward skew in the output probabilities, with the left end of the region being sparsely populated and the rightward end being densely populated. Sharpness, which is the desire for models to always predict with high confidence, i.e., predicted probabilities concentrate to 0 or 1, is a fundamental property (Gneiting et al., 2007). (That is, network predictions are typically very confident.) This causes only a few bins to contribute the most to ECE—typically one or two as bin sizes are 10-20 in practice (Guo et al., 2017).

### 3.3 BIAS-VARIANCE TRADEOFF

'Ground truth' is the calibration error we would measure if we could see an infinite number of predictions at a particular confidence level. That would give us a long-run error for every prediction, which we could aggregate into an overall calibration error. Because we don't have that infinite set of reference predictions for every real-valued prediction, we try to approximate that set by using the values that are close together as reference predictions. As we increase the number of datapoints in a bin, we get a more accurate estimate of the calibration in a given range through the added (seen as a reduction in variance). But as the datapoints in the bin start to differ from one another due to the extension of the range, they bias the estimate of the calibration within that bin (we refer to this as bias).

Selecting the number of bins has a bias-variance tradeoff as it determines how many data points fall into each bin and therefore the quality of the estimate of calibration from that bin's range. In particular, a larger number of bins causes more granular measures of calibration error (low bias) but also a high variance of each bin's measurement as bins become sparsely populated. This tradeoff

compounds particularly with the problem of fixed calibration ranges, as due to sharpness certain bins have many more data points than others.

### 3.4 PATHOLOGIES IN STATIC BINNING SCHEMES

Metrics that depend on static binning schemes like ECE suffer from issues where you can get near 0 calibration error due to overconfident and underconfident predictions overlapping in the same bin. These cancellation effects may not always be extreme enough to push an incredibly uncalibrated model to 0 calibration error, but it can be difficult to judge when using a static metric whether differences between models or techniques are due to a true improvement in the model's calibration or whether improved calibration is due to cancellation.

Code in the appendix (Figure 5) demonstrates this effect.

## 4 CHALLENGES IN CALIBRATION

Before describing new metrics for calibration, we first outline broad challenges with designing such metrics.

### 4.1 GROUND TRUTH & COMPARING CALIBRATION METRICS

There are many challenges in measuring a network's calibration, beginning with the absence of ground truth. In principle, one can limit comparisons to controlled, simulated experiments where ground truth is available by drawing infinitely many samples from the true data distribution. However, even with ground truth, any estimation property such as bias remains difficult to compare across estimators, as "ground truth error" is multi-valued and estimators may make measurements for different elements of these values. Specifically, calibration is a guarantee for all predicted probabilities $p \in [0, 1]$ and class labels $y \in \{0, \ldots, K - 1\}$ (see also the calibration function of Vaicenavicius et al. (2019)). An estimator may have lower bias and/or variance in estimating the error for specific ranges of $p$ and $y$ but higher bias and/or variance in other ranges.

The other major challenge with respect to ground truth is that due to the differences in their assumptions, different calibration metrics approximate different overall calibration values. ECE doesn't look at all predictions, and is estimating a different value than SCE. The ACE vs SCE comparison is possible with respect to the probabilities they look at, but as the static binning scheme imposes a different bias than the adaptive scheme, it's a challenge to compare static metrics to adaptive metrics.

### 4.2 WEIGHTING

Because calibration error is multi-valued, and a desirable metric is scalar-valued, the question of how to weight probability values (where one can see thresholding as a 0/1 weighting on datapoints below / above the threshold) creates a set of differences in calibration error metrics that choose to emphasize different aspects of calibration performance. In many contexts what matters most is the rare event—the network's classifications leading up to an accident, the presence of a planet, or the presence of a rare disease. Knowing the difference between whether a class's true probability is .01 and .001 (the difference between 1 in 100 and 1 in 1000) is both extremely difficult to discern and may be much more relevant than a difference between .3 and .301, which these calibration metrics would treat as equivalent. In these contexts, weighting the ends of the interval close to 0 and 1 would be ideal.

Additionally, in the context of out-of-distribution detection, we would prefer to be well-calibrated in the middle of the spectrum, whereby a prediction of .5 (high uncertainty) is more likely to happen.

## 5 NEW CALIBRATION METRICS

Each calibration error metric is evaluated on each class, independently. The results for each class are averaged.

## 5.1 MULTICLASS & STATIC CALIBRATION ERROR

We first introduce Static Calibration Error (SCE), which is a simple extension of Expected Calibration Error to every probability in the multiclass setting. SCE bins predictions separately for each class probability, computes the calibration error within the bin, and averages across bins:

$$\text{SCE} = \frac{1}{K} \sum_{k=1}^{K} \sum_{b=1}^{B} \frac{n_{bk}}{N} \left| \text{acc}(b, k) - \text{conf}(b, k) \right|.$$

Here, $\text{acc}(b, k)$ and $\text{conf}(b, k)$ are the accuracy and confidence of bin $b$ for class label $k$, respectively; $n_{bk}$ is the number of predictions in bin $b$ for class label $k$; and $N$ is the total number of data points. Unlike ECE, assuming infinite data and infinite bins, SCE is guaranteed to be zero if only if the model is calibrated.

## 5.2 ADAPTIVITY & ADAPTIVE CALIBRATION ERROR

Adaptive calibration ranges are motivated by the bias-variance tradeoff in the choice of ranges, suggesting that in order to get the best estimate of the overall calibration error the metric should focus on the regions where the predictions are made (and focus less on regions with few predictions). This leads us to introduce Adaptive Calibration Error (ACE). uses an adaptive scheme which spaces the bin intervals so that each contains an equal number of predictions.

In detail, ACE takes as input the predictions $P$ (usually out of a softmax), correct labels, and a number of ranges $R$.

$$\text{ACE} = \frac{1}{KR} \sum_{k=1}^{K} \sum_{r=1}^{R} \left| \text{acc}(r, k) - \text{conf}(r, k) \right|.$$

Here, $\text{acc}(r, k)$ and $\text{conf}(r, k)$ are the accuracy and confidence of adaptive calibration range $r$ for class label $k$, respectively; and $N$ is the total number of data points. Calibration range $r$ defined by the $\lfloor N/R \rfloor$th index of the sorted and thresholded predictions.

### 5.2.1 SHARPNESS & ADAPTIVE CALIBRATION ERROR

A major motivation for the Adaptive Calibration Error was the realization that the vast majority of predictions made by deep learning models are close to 1 or 0. A static binning scheme will group almost all predictions into two bins, one at each end, with very sparse predictions falling in the bins that are in between (Figure 3, Left). ACE adapts to focus on the areas where predictions are densest (Figure 3, Right).

# 6 INTERPRETABILITY & ANALYSIS

## 6.1 INTERPRETATION OF CALIBRATION ERROR METRICS

The natural interpretation for both metrics that account for every prediction made by a classification model (SCE and ACE) is that the calibration error is the average error of all predictions made by the mode. The natural interpretation for ECE, in contrast, is that its calibration error is the average error of the highest confidence prediction made by the model.

## 6.2 CRITICISMS & LIMITATIONS OF ADAPTIVE CALIBRATION METRICS

These adaptive calibration metrics can create excessively large ranges if there is very sparse output in a region. In those cases, it may be better to have a higher variance estimate of the calibration, but only compare datapoints that are closer together (dropping datapoints that are farther away). This level of granularity can be replicated with an adaptive scheme that has a very large number of ranges, but that will come at the cost of easy interpretability and will increase the within-range variance across all ranges, not just the sparse regions.

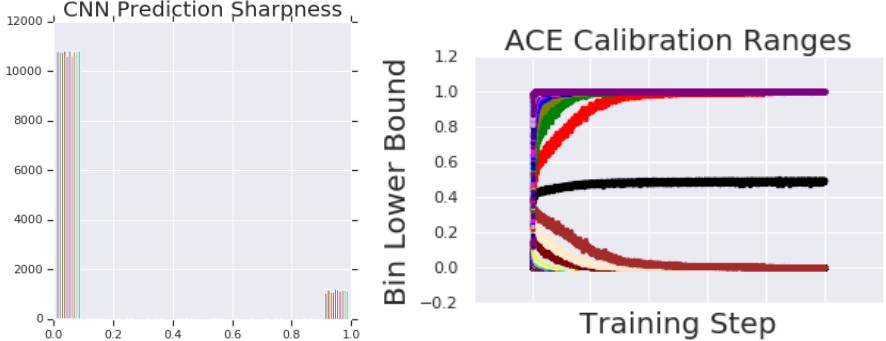

Figure 3: **Left:** Model sharpness, where the vast majority of confidence scores are very near 1 or very near 0, motivates ACE. This class-conditional histogram of confidence scores on 12,000 samples from Fashion-MNIST demonstrates that sharpness. **Right:** Binning scheme chosen by ACE measured at each step of training on Fashion-MNIST. Each bin measuring the calibraion error of datapoints within a particular range has a lower bound to the bin and an upper bound. As the model's predictions become sharper, the chosen bins adapt to focus on regions with a high concentration the predictions. Each color represents the lower bound of the nth largest bin, for 100 bins.

This scheme does not accomplish transfer smoothly between all related predictions, although en-sembling / hierarchical calibration ranges could approximate that smoothness. This limitation does suggest an smoother metric, akin to our algorithm that leverages dynamic programming to look at very many range schemes simultaneously & efficiently.

With a static bin calibration metric it can be straightforward to compare the outputs of two very different models through a reliability diagram. Due to the variability of their binning scheme based on the density of the model's predictions, adaptive calibration metrics make it more difficult to compare two models side-by-side in a reliability diagram, as their points fall in different regions depending on the bin locations that are selected.

### 6.3 Hyperparameter Analysis

Adaptive calibration metrics are more sensitive to the binning hyperparameter. Empirically, there are larger changes in calibration error due to bin count choice than static metrics. This variance decreases as the bin count rises, leveling off at 50 in our experiments. We suggest using ACE with 50+ bins for models trained on datasets akin to CIFAR or ImageNet.

## 7 Experiments

### 7.1 Recalibration Experiments

We trained a ResNet-110 model (He et al., 2016) on CIFAR-100 (Krizhevsky & Hinton, 2009). We then estimated the regression parameters, for each postprocessing method, on the validation held-out set. Finally, we used the parameter estimates to compute calibrated probabilities on the test set. The results are reported in Table 3. The same experiment with a ResNet-50 model on ImageNet 2012 is reported in Table 4 and echoes the findings of Table 3.

Table 3 shows that temperature scaling outperforms the other 3 methods on ECE as reported in (Guo et al., 2017). However, temperature scaling falls short in comparison to vector scaling on the SCE and ACE metrics. In fact, the relative gap on SCE is the largest, among all metrics and across methods. This is unsurprising given that SCE is a multiclass extension of ECE which considers all probabilities, instead of the top one. Vector scaling typically does best.

For example, temperature scaling (Guo et al., 2017) has been shown to effectively minimize expected calibration error better than alternative techniques such as isotonic regression and Platt scaling

| Method | ECE | SCE | ACE |
|---|---|---|---|
| Uncalibrated | 19.64% | 0.41% | 0.13% |
| Temp. Scaling | **2.16%** | 0.06% | 0.06% |
| Vector Scaling | 2.27% | **0.04%** | **0.01%** |
| Matrix Scaling | 12.11% | 0.26% | 0.26% |
| Isotonic Regr. | 17.85% | 0.35% | 0.12% |

Table 1: ECE, TACE, SCE, and ACE (with 15 bins) on a ResNet-110 applied to CIFAR-100 before calibration, and after the application of post-processing methods. The best recalibration method depends on the metric, which motivates its study.

| Method | ECE | SCE | ACE |
|---|---|---|---|
| Uncalibrated | 6.63% | 0.02% | 0.015% |
| Temp. Scaling | 5.42% | 0.01% | 0.001% |
| Vector Scaling | **1.44%** | **0.002%** | 0.004% |
| Matrix Scaling | 5.06% | 0.01% | 0.001% |
| Isotonic Regr. | 3.474% | 0.01% | **0.000%** |

Table 2: ECE, SCE, and ACE (with 15 bins) on a ResNet-50 applied to ImageNet before calibration, and after the application of various extensions to Platt scaling and Isotonic regression. These percentages across metrics are not directly comparable.

(Platt et al., 1999). The question of whether these scaling techniques effectively minimize more sophisticated error metrics is an important standard for their efficacy.

In fact, by design of having only a single scalar parameter which uniformly scales predicted probabilities across all classes, temperature scaling is likely perform worse when accounting for, for example, calibration errors across all predictions. We hypothesize that this would be the case with respect to adaptive calibration metrics (e.g. ACE) as well as calibration metrics that account for the probabilities of all classes instead of that of only the top predicted class (e.g. SCE).

When designing post-processing methods, not accounting for the properties detailed in Section 3 during evaluation can lead to misleading conclusions about a post-processing method's success.

## 7.2 LABEL NOISE

We train softmax regression (multinomial logistic regression) models at 40 levels of label noise from 0.0 to 0.05. The label noise value corresponds to the percentage of labels that are randomly assigned, where the correct label is included as one of the possible random assignments.

One way to measure the importance of a sample is its confidence. We look at the fraction of important predictions ECE omits by setting a threshold confidence (.01 in our visualized example, see Figure 1 Top Right) and observe that this fraction rises with the difficulty of the modeling task (measured by model accuracy and mean maximum confidence, see Figure 1 Bottom). This result holds for a wide range threshold levels.

## 8 CONCLUSION

This paper studies metrics for evaluating calibration. We makes the following contributions:

1. We identify problems with the widely used ECE metric to assess calibration.
2. We propose new metrics (SCE and ACE) to address each of these problems.
3. We recommend best practices for how to evaluate calibration using our metrics.

We believe that the way neural network uncertainties are evaluated in future must be aware of the challenges we raise, and in light of those challenges recommend the use of SCE or ACE over ECE for evaluating a multiclass classifier's calibration error.

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

| Method | ECE | TACE | SCE | ACE |
|---|---|---|---|---|
| Uncalibrated | 19.64% | 10.03% | 0.41% | 0.13% |
| Temp. Scaling | **2.16%** | **0.52%** | 0.06% | 0.06% |
| Vector Scaling | 2.27% | 0.61% | **0.04%** | **0.01%** |
| Matrix Scaling | 12.11% | 3.98% | 0.26% | 0.26% |
| Isotonic Regr. | 17.85% | 2.83% | 0.35% | 0.12% |

Table 3: ECE, TACE, SCE, and ACE (with 15 bins) on a ResNet-110 applied to CIFAR-100 before calibration, and after the application of post-processing methods. The best recalibration method depends on the metric, which motivates its study.

| Method | ECE | TACE | SCE | ACE |
|---|---|---|---|---|
| Uncalibrated | 6.63% | 2.51% | 0.02% | 0.015% |
| Temp. Scaling | 5.42% | 2.64% | 0.01% | 0.001% |
| Vector Scaling | **1.44%** | **1.25%** | **0.002%** | 0.004% |
| Matrix Scaling | 5.06% | 1.98% | 0.01% | 0.001% |
| Isotonic Regr. | 3.474% | 1.862% | 0.01% | **0.000%** |

Table 4: ECE, TACE, SCE, and ACE (with 15 bins) on a ResNet-50 applied to ImageNet before calibration, and after the application of various extensions to Platt scaling and Isotonic regression. These percentages across metrics are not directly comparable.

Maithra Raghu, Katy Blumer, Rory Sayres, Ziad Obermeyer, Robert Kleinberg, Sendhil Mullainathan, and Jon Kleinberg. Direct uncertainty prediction for medical second opinions. 2018.

Seonguk Seo, Paul Hongsuck Seo, and Bohyung Han. Learning for single-shot confidence calibration in deep neural networks through stochastic inferences. In *Proceedings of the IEEE Conference on Computer Vision and Pattern Recognition*, pp. 9030–9038, 2019.

Sunil Thulasidasan, Gopinath Chennupati, Jeff Bilmes, Tanmoy Bhattacharya, and Sarah Michalak. On mixup training: Improved calibration and predictive uncertainty for deep neural networks. *arXiv preprint arXiv:1905.11001*, 2019.

Athanasios Tsoukalas, Timothy Albertson, and Ilias Tagkopoulos. From Data to Optimal Decision Making: A Data-Driven, Probabilistic Machine Learning Approach to Decision Support for Patients With Sepsis. *JMIR Medical Informatics*, 3(1):e11, February 2015. ISSN 2291-9694. doi: 10.2196/medinform.3445. URL http://medinform.jmir.org/2015/1/e11/.

Juozas Vaicenavicius, David Widmann, Carl Andersson, Fredrik Lindsten, Jacob Roll, and Thomas B Schön. Evaluating model calibration in classification. *arXiv preprint arXiv:1902.06977*, 2019.

Bianca Zadrozny and Charles Elkan. Transforming classifier scores into accurate multiclass probability estimates. In *Proceedings of the eighth ACM SIGKDD international conference on Knowledge discovery and data mining*, pp. 694–699. ACM, 2002.

## A  APPENDIX

### A.1  THRESHOLDING & THRESHOLDED ADAPTIVE CALIBRATION ERROR

One initial challenge is that the vast majority of softmax predictions become infinitesimal (Figure 3, Top Left). These tiny predictions can wash out the calibration score, especially in the case where there are many classes, where a large proportion of them model's predictions correspond to an incorrect class. One response is to only evaluate on values above a threshold $\epsilon$. Mathematically, TACE is identical to ACE, with the only difference being that TACE is only evaluated on values above $\epsilon$. These predictions overlap with the predictions evaluated by ECE (all maximum values per datapoint), leading them to have similar reactions to recalibration methods.

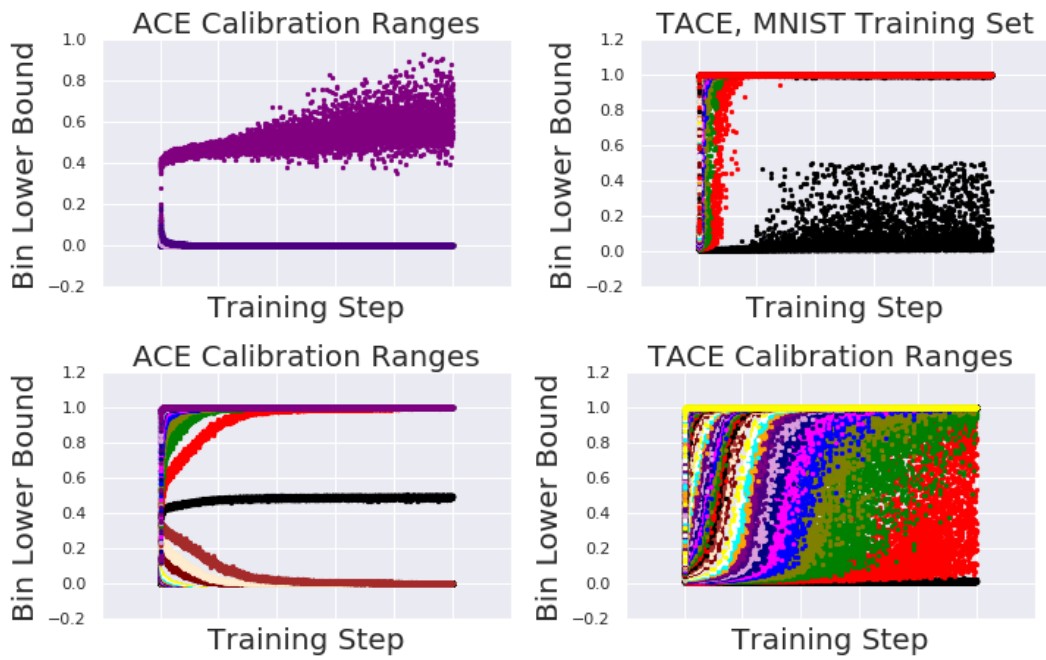

Figure 4: **Top Left:** Lower bounds of calibrations ranges over the course of training for adaptive calibration error on Fashion-MNIST, focusing almost entirely on small ranges and motivating thresholding. **Top Right:** On the MNIST training set with thresholding, so few values are small that the bottom of the lowest range often spikes to .99 and higher due to every datapoint being fit. **Bottom Left:** ACE on Fashion-MNIST validation with 100 calibration ranges. **Bottom Right:** Thresholded adaptive calibration with 50 calibration ranges over the course of training on Fashion-MNIST's validation set.

```python
predictions = []
correct_values = []
for i in range(450):
  value = .48
  predictions.append([1-value, value])
  correct_values.append(1)
for i in range(550):
  value = .42
  predictions.append([1-value, value])
  correct_values.append(0)
softmax = np.array(predictions)
print(expected_calibration_error(softmax, correct_values, n_bins=10))
```

Figure 5: This python code (with numpy imported as np, and with an expected calibration error function loaded in the environment) implements a pathology in ECE which will return 0 calibration error by including overconfident and underconfident values in the same bin. The softmax here has 450 predictions at .52 (ECE only looks at the maximum value) and evaluates all of them to be incorrect, despite being predicted with high confidence. The softmax also has 550 predictions at .58, all evaluated to be correct, though they're predicted with low confidence (relative to 1 for correct and 0 for incorrect). Despite this extreme miscalibration, the expected calibration error is exremely low ( 0.003).

