# OpenReview forum: "Measuring Calibration in Deep Learning"
_ICLR.cc/2020/Conference — Reject_

### Official Review · AnonReviewer2 · 2019-10-20
**Official Blind Review #2**

**Rating:** 6

**Review:**

After an interesting review of calibration methods, the paper describes two new methods for assessing calibration. The first method, SCE, is an extension of the usual ECE to the multi class setting. The second method, ACE, is a slight variation where bins are computed adaptively.

The paper is interesting and relatively well written. Although the contribution is rather simple (can be describe in less than a page), I can see myself using the SCE/ACE metrics in the near future.

In order to strengthen the submission, I still feel that the authors should try to describe why the newly introduced metrics can help in applied scenarios.
(1) OK, the ECE omits a lot of important predictions (e.g. fig 1) --> give a real application where this matters
(2) OK, adaptive binning seems a sensible approach. --> give a real application where a difference between 0.99 and 0.999 does make a difference

The newly introduced metrics *are* interesting, and the theoretical justifications *are* sensible. The paper would be even better if the applied motivations were better described. Since the proposed method does have the potential to be applied in industrial/applied scenarios, it is slightly disappointing that it is presented as another academic exercise.

minor remark: it is not clear why the factor (1/K) is needed in the definition of SCE since the weights are already summing up to one -- this makes SCE comparisons between datasets with different number fo classes more difficult.

**Experience Assessment:**

I have read many papers in this area.

**Review Assessment: Checking Correctness Of Derivations And Theory:**

I carefully checked the derivations and theory.

**Review Assessment: Checking Correctness Of Experiments:**

I carefully checked the experiments.

**Review Assessment: Thoroughness In Paper Reading:**

I read the paper thoroughly.

---

### Official Review · AnonReviewer3 · 2019-10-21
**Official Blind Review #3**

**Rating:** 3

**Review:**

Summary:
	This paper proposes two novel scalar-valued evaluation measures, namely Static Calibration Error (SCE) and Adaptive Calibration Error (ACE)  for evaluating the quality of calibration (the accuracy of estimating p(y|x) in classification) in deep learning. The authors suggested that the existing Expected Calibration Error (ECE), which is the current most popular scalar-valued metric for calibration, has several drawbacks, especially in the multi-class scenario. Intuitively, ECE only focuses on the predicted class probability (argmax) and ignores class probabilities of other K-1 classes, which implies that ECE may fail to capture the class probability estimates for all classes. They also illustrated the drawback of ECE under label noise.

========================================================
Contribution:
1. Pointing out that ECE has several drawbacks in multiclass scenario, e.g., does not take predictions of all classes but only the one with maximum confidence.
2. Proposing two novel measures: SCE and ACE, where SCE is a natural extension to make sure the metric consider all class probability estimates, and ACE is adaptive in the sense that it is focus on the regions where many predictions are made.
3. Conducting experiments to illustrate that although temperature scaling may work very well when we used ECE as a metric, vector scaling can be advantegous when we consider SCE or ACE.

Clarity:
Although there are typos, it is not difficult to understand the motivation and what this paper is trying to propose. But it is suggested that the paper was done in a rush manner.

1. Issue in Figure 1 and experiment 7.2:

I found that Figure 1 is difficult to understand and I may misunderstand. Moreover, I couldn't get the main message of it. 		1.1 (Top-left), the message I got is that the error based on each metric is bigger as the label noise increases (but each metric is incomparable).
1.2 (Top-right), I couldn't get what Predictions ECE omits over threshold 0.1 means. It would be better to clearly explain it, e.g., how to compute % ECE omits over a certain threshold.
1.3 (Bottom), I learned that with label noise, both accuracy and model confidence (which I think is max p(y|x) decreases as the noise increases, which is common. Moreover, does Against and vs. mean the same thing in this context, then using the same word can make it more consistent.

	Moreover, I wonder why we have to focus on the label noise because even in the normal scenario, ECE should have drawbacks already too and it is more interesting for me to see the illustration in the normal scenario. On the other hand, if I don't misunderstand, ECE did not omit a lot of predictions according to the top-right figure if there is no label noise. In practice, we may use a more sophisticated method to handle label noise. Because under label noise, the class probability estimation is already incorrect theoretically, i.e., it may shift depending on the noise type (Menon+, ICML2015: Learning from Corrupted Binary Labels via Class-Probability Estimation). And I am not sure why we have "(Figure 1)" at the end of the first sentence of Sec. 3.1, is that sentence related to Figure 1? In my opinion, Figure 2 is much better to visualize that ACE may capture things that ECE fails to capture. Finally, Figure 1 is a part of experiment 7.2 and I think it is fine to move this to the experiment section as an additional experiment under the label noise scenario. Finally, how to train your model in Figure 1, is it uncalibrated version? i.e., without temperature scaling or other modifications.

2. Figure 3 is difficult to understand.
	2.1 (Left) what is sharpness, how to calculate sharpness and what is the y-axis? And what is the x-axis here, is it a confidence score?.
	2.2 (Right) What is the training step? And there is no value specified in the x-axis. I couldn't understand how to plot this figure.
3. Tables 1 and 2 are never discussed and in "Table 1: ECE, TACE, SCE, and ACE", there is no TACE and never mention in the main body of the paper. Instead, the main body discussed about Tables 3 and 4, which is not in the main body (or is it? since it is in between the reference).
4. How many trials did you run the experiment and what criteria you use to give boldface to a method? Since this paper also highly relies on the experimental results, it would be great to clarify.

========================================================
Comments:
The authors did a great job to point out the problem of ECE. Although SCE is a very simple and natural extension of ECE, its contribution is significant because it relevates the drawback of ECE as the authors suggested. I believe this work can make an impact to the field. For ACE, I have an impression that it is difficult to use. Also, it would be nice to see the performance with respect to Maximum Calibration Error (MCE), which is completely ignored in this paper. Because in MCE, we can see that temperature scaling did not almost always perform significantly better than other methods as it performed with respect to ECE (in Appendix of Guo et al., 2017), which is similar to what we observed in SCE and ACE.

Unfortunately, although I like the idea of this paper, I found that the clarity of the paper is insufficient in its current state. It seems that the paper was really done in a rush and thus the writing can be highly improved. As a result, given the current manuscript, I vote a weak reject for this paper.

========================================================
Additional questions:
1. Why the name of SCE is static calibration error? If you mean it is not adaptive as ACE, then ECE is also static in this sense. Therefore, it may be a good idea to come up with a different name,e.g., classwise calibration error.
2. May SCE and ACE suffer from class-imbalance scenario more than ECE?
3. Are there any advantages of ECE over SCE and ACE?

========================================================
Potential typos I found:
1. Abstract: the last sentence: taks -> takes
2. Abstract: Overconf. and underconf. is -> are
2. INTRO: the first sentence of the last paragraph may algorithms -> many algorithms
3. INTRO: 4th paragraph: has lead -> has led
4. INTRO: last sentence: Static Calibratinon -> Static Calibration
5. INTRO: last sentence Adaptive Calibration -> Adaptive Calibration Error
6. 2.1: {(x,y)} should be {(x,y)}_{i=1}^{N}?
7. all predictions made by the mode -> all predictions made by the model
8. in fig.3: calibraion -> calibration
9. Table 1: remove TACE



**Experience Assessment:**

I have published one or two papers in this area.

**Review Assessment: Checking Correctness Of Derivations And Theory:**

I assessed the sensibility of the derivations and theory.

**Review Assessment: Checking Correctness Of Experiments:**

I carefully checked the experiments.

**Review Assessment: Thoroughness In Paper Reading:**

I read the paper thoroughly.

---

### Official Review · AnonReviewer1 · 2019-10-27
**Official Blind Review #1**

**Rating:** 1

**Review:**

Paper summary: This paper proposes two simple extensions to Expected Calibration Error (ECE):  1) SCE which accounts for multiclass classification settings by averaging over all the errors due to all classes (as opposed to error in the top one class only in ECE) and 2) ACE which attempts as distributing the predictions equally across the bins (as opposed to have too few highly populated bins in the interval). Authors evaluated their approach using ResNet-110 on CIFAR100 and ResNet-50 on ImageNet against most common post-training calibration methods.

Pros:
(+): The paper is well-motivated.
(+): The problem is important and has direct real world applications.
(+): The idea is simple and viable to improve ECE.

Cons that significantly affected my score and resulted in rejecting the paper are as follows:

1 - Experimental setting and evaluations:
The biggest drawback in this paper is the experimental setting which is not rigorous enough to show the effectiveness of the proposed metrics due to the following reasons:
(a) The first proposed extension (SCE) is too incremental. However, while I am not against simple and effective methods (I have even listed this as a pro above), I think it should be backed up with more thorough experiments and discussions. SCE is supposed to be more effective in case of having more number of classes however authors do not shed light into this properly by taking advantage of comparing their CIFAR100 and ImageNet 1K experiments. They keep emphasizing on the fact that the numbers provided in Table 1 and 2 are not comparable so how is the reader supposed to understand their differences? They also show their metrics give far less error to models than ECE in Table 1 and 2 and yet it is not clear why that is. Not to mention that authors need to state these results are obtained on how many runs and to report stds because the numbers appear too close and are hence not conclusive.
(b) Datasets and architectures: Authors have used MNIST and FASHION-MNIST when discussing the shortcomings of ECE. I was wondering what architecture they used for these experiments? The reason I am asking is that it is a known fact (see Guo et al., ICML17) that the simpler the architecture is, the more calibrated its predictions tend to be and vice versa. For instance LeNet 5 appears to be much more well-calibrated than more modern neural networks such as ResNet variants despite being less accurate (Guo et al, 2017). Therefore, it will probably be in their favor to show these issues on a less calibrated model.
(c) Can authors please explain why they have used ResNet-110 on CIFAR100 and ResNet-50 on ImageNet? I assume it might be because they intended to compare to (Guo et al, 2017) but there are more experiments there for comparison. Have they also tried any other architecture? It has been shown before that there can be a noticeable difference across different architectures used on a fixed dataset. Authors may want to add the arch effect to their evaluation.

2) The second metric (ACE) while it is well-motivated, it is defined such that it leaves the impression that applying it will involve lots of heuristics as there is no systematic procedure is given. In section 6.3 (where I think could be a section to address the implementation details for this) no effective information is given either expect that authors recommend using 50+ bins for CIFAR and ImageNet without any quantitative support. It is also not clear why authors used 15 bins in their experiments for these datasets in Table 1 and 2 (maybe comparing to the baseline? but they could show both specially if 50+ bins is better for ACE). Moreover, in section 5.2 where it says “the overall calibration error the metric should focus on the regions where the predictions are made (and focus less on regions with few predictions).“, I was wondering if authors have any suggestions on how to identify the predictions with low confidence as coming from in or out-distribution?

2- Structure of the paper, writing, and visualizations:

(a) Writing: The paper, in its current form, needs to be thoroughly proofread and reorganized. The text does not read well and is vague in most parts. Authors have spent too much time explaining the drawbacks of the prior work in 4 pages (entire sections 2,3, and 4) only to propose their ideas in section 5.1 and 5.2 in almost half a page. Unfortunately the analysis section (6) acts very poorly in providing a thorough exploration into their method.

(b) Dividing each section to too many subsections has hurt the flow as each section is too short and does not provide deep evidence to support its title/subtitle.

<<< Note that the followings are less major and are given only to help, and not necessarily part of my decision assessment >>>

(c) Results shown in Table 3 and 4 are reporting “TACE” which is technically explained in the Appendix. Apart from the fact that authors need to make sure the Table comes before the references (not in the middle) but more importantly, as a paper submitted to ICLR I would expect it to be self-contained and be able to provide all the details needed. Authors should either move this Table to the appendix or move section (A.1) to the main text.

(d) In section 3.4. Authors provide support for their claim on issues with binning scheme with a screenshot of their code in the Appendix. I think it is important to show this effect but in perhaps a table with quantitative results within the main text.

(e) Title for Section 5 as the main contribution of this paper should be changed to reflect that.

(f) Abstract in its current wording, does not provide absolute no detail into the proposed metrics. Whereas it can, because 1) it encourages the reader to keep reading 2) the method is simple enough to be summarized here.

(g) The figures do not meet the conventional scientific standards and have to be significantly improved. Standard deviations are missing.

(h) There are also grammar errors and typos, (for example on page 1 paragraph 5, the word "may" should be "many"), for which I have found passing my writing through the free version of Grammarly very helpful in getting rid of most such errors.

As a final note, I think this paper can be contributive to the field as it provides a novel and simple extension to a widely used calibration metric. However it needs to be written in a more effective way and be supported by a more rigorous experimental setting. Therefore, I will be willing to change my score if the presented issues will be addressed.


**Experience Assessment:**

I have read many papers in this area.

**Review Assessment: Checking Correctness Of Derivations And Theory:**

I carefully checked the derivations and theory.

**Review Assessment: Checking Correctness Of Experiments:**

I carefully checked the experiments.

**Review Assessment: Thoroughness In Paper Reading:**

I read the paper thoroughly.

---

### Public Comment · ~Yukun_Ding1 · 2019-09-28
**A related work**

Great work! The discussion on the calibration in the multiclass setting is very informative. You may want to cite the following paper that focuses on a similar problem.
https://arxiv.org/abs/1903.02050

---

### Public Comment · ~Xinshao_Wang1 · 2022-10-16
**Great work. For a sharing and discussion purpose about calibration error, we propose a Generic coarse Signed Calibration Error (GSCE).**

In a recent work, named [ProSelfLC: Progressive Self Label Correction Towards A Low-Temperature Entropy State](https://arxiv.org/abs/2207.00118),
we have
* a technical subsection 4.1 about calibration error, where the Generic coarse Signed Calibration Error (GSCE) is proposed.
* an empirical analysis subsection 4.2 to visualize the miscalibration.

---

### Decision · Program_Chairs · 2019-12-19

**Decision:**

Reject

**Comment:**

The authors propose two measures of calibration that don't simply rely on the top prediction. The reviewers gave a lot of useful feedback. Unfortunately, the authors didn't respond.